# Tight Interplay between Replication Stress and Competence Induction in *Streptococcus pneumoniae*

**DOI:** 10.3390/cells10081938

**Published:** 2021-07-30

**Authors:** Vanessa Khemici, Marc Prudhomme, Patrice Polard

**Affiliations:** 1Laboratoire de Microbiologie et Génétique Moléculaires (LMGM), Centre de Biologie Integrative (CBI), Centre National de la Recherche Scientifique (CNRS), 31062 Toulouse, France; vkemici@gmail.com (V.K.); marc.prudhomme@univ-tlse3.fr (M.P.); 2Université de Toulouse, Université Paul Sabatier, 31062 Toulouse, France

**Keywords:** DNA damage response, genome integrity, replication stress, recombinational repair, bacterial competence, *Streptococcus pneumoniae*

## Abstract

Cells respond to genome damage by inducing restorative programs, typified by the SOS response of *Escherichia coli*. *Streptococcus pneumoniae* (the pneumococcus), with no equivalent to the SOS system, induces the genetic program of competence in response to many types of stress, including genotoxic drugs. The pneumococcal competence regulon is controlled by the origin-proximal, auto-inducible *com*CDE operon. It was previously proposed that replication stress induces competence through continued initiation of replication in cells with arrested forks, thereby increasing the relative *com*CDE gene dosage and expression and accelerating the onset of competence. We have further investigated competence induction by genome stress. We find that absence of RecA recombinase stimulates competence induction, in contrast to SOS response, and that double-strand break repair (RexB) and gap repair (RecO, RecR) initiation effectors confer a similar effect, implying that recombinational repair removes competence induction signals. Failure of replication forks provoked by titrating PolC polymerase with the base analogue HPUra, over-supplying DnaA initiator, or under-supplying DnaE polymerase or DnaC helicase stimulated competence induction. This induction was not correlated with concurrent changes in origin-proximal gene dosage. Our results point to arrested and unrepaired replication forks, rather than increased *com*CDE dosage, as a basic trigger of pneumococcal competence.

## 1. Introduction

All genomes are susceptible to potentially life-threatening damage, of both environmental and endogenous origin. To confront this challenge to their survival, organisms have evolved a range of repair pathways, notably those inducible by the damage itself. The first program found to be triggered in response to DNA damage was the SOS response of *Escherichia coli* [1,2]. It is set in motion when some mishap generates single-stranded DNA on which the RecA protein polymerizes; the resulting nucleofilament activates autoproteolysis of the transcriptional repressor LexA and thus expression of the LexA-controlled genes that provide DNA repair functions.

The SOS system is widespread in bacteria but not universal, and some species have evolved other responses to genome damage [3]. A prominent example is *Streptococcus pneumoniae* (the pneumococcus), which does not encode a LexA homologue but responds to genotoxic agents by developing the distinct physiological state of competence [4]. Competence provides the cells with new properties, including the well-known horizontal gene transfer process of genetic transformation [5]. Many species share this ability to convert to competence for transformation, as well as a similarity of mechanism in the importing of external DNA across the cell membrane and integration into the genome by RecA-directed homologous recombination [5]. On the other hand, regulation of competence appears specific to each species, reflecting the distinct integration of the genetic program into each bacterial lifestyle [6]. Here, we report our analysis of pneumococcal competence induction in response to various types of genome stress that disrupt DNA recombination and replication dynamics. 

The core of pneumococcal competence control is a two-component system (TCS) consisting of the histidine kinase ComD and the transcription regulator ComE [6,7]. The ComDE TCS is modulated by a competence-stimulating peptide (CSP), encoded by the *comC* gene in the form of a pre-CSP that is matured and exported through the cell membrane by the ABC transporter ComAB. These elements define a positive feedback loop, termed the ComABCDE core sensing system, which integrates various kinds of stress and coordinates development of competence in the population (Figure 1A; [8]). In liquid cultures, competence induction occurs only during the exponential phase of growth and after a time delay characteristic of the strain and growth conditions (a ‘*pre-competence*’ period referred to as X_A_; Figure 1A [9]). Induction of competence results in two successive transcriptional waves, termed *early* and *late*. The early wave is initiated by ComE upon its phosphorylation by ComD [8,10]. Phosphorylated ComE activates about 20 early competence (*com*) genes, including *com*X whose product, the sigma factor σ^X^, primes the late wave by governing the transcription of about 60 other *com* genes. The σ^X^ regulon includes DprA that, in addition to mediating RecA-directed recombination during transformation [11], interacts with phosphorylated ComE to shut off competence [12]. Thus, the shift to competence as measured by *com* gene expression is transient, lasting for roughly one generation (referred to as the X_B_ period [9]; Figure 1A). A feature of pneumococcal competence is that it develops only during the exponential phase of growth, during which the replicative state of the chromosome renders it particularly sensitive to damage or disruption. 

A previous study showed pneumococcal competence to be induced by drugs that alter genome integrity or replication [13]. Competence was proposed to substitute for the SOS system in the pneumococcus, which does not possess a LexA homologue [4,14]. This raised the question of how replication stress signals are conveyed to the ComDE-mediated competence induction system. It was observed that these stresses were correlated with increased abundance of transcripts originating from the *ori*-proximal *com*CDE operon whose gene dosage had risen as a result of deceleration of the replication forks [15]. Here, we have further studied the signaling that triggers competence development following genome damage in the pneumococcus. We first found that RecA and the main homologous recombination pathways RexAB and RecFOR are not involved in this signaling. Rather, RecA prevents competence development via its recombinational repair activity. While we also observed a link between fork arrest and competence development, our results do not support the notion that signaling stems from a *com*CDE gene dosage effect. Instead, they suggest that arrested and unrepaired replication forks are the trigger for competence induction. 

## 2. Material and Methods

### 2.1. Pneumococcal Strains, Cell Growth, Competence Recording, and Transformation Procedures

*Streptococcus pneumoniae* strains used in this study are listed in Appendix A. They are all derivatives of R800, which derives from the unencapsulated R6 strain originating from D39 lineage [9]. 

All strains were stored at −70 °C in the form of stock cultures, which were used to inoculate liquid fresh medium to measure competence development under the distinct conditions described in the text and/or the figures legends. Stock cultures were prepared at 37 °C in liquid C+Y medium [16] at pH 6.8 to impede spontaneous competence development. Cells were grown to OD_550_ = 0.2, centrifuged at 6000 rpm for 5 min at 4 °C to discard the medium, resuspended with fresh C+Y medium containing 15% glycerol to OD_550_ = 0.4, and kept frozen at −70 °C. In the case of ^e^P*_lac_*::*dnaA*, ^e^P*_lac_*::*dnaE,* and ^e^P*_lac_*::*dnaC* strains, C+Y medium was supplemented with 10 µM IPTG, which is the concentration that results in a WT growth rate and *ori*/*ter* ratio value. 

Competence was monitored in real time over the growth of the cell population by using a transcriptional fusion between the *luc* firefly luciferase gene and the *com*C promoter (in the case of the *rec*A^−^ and *rex*B^−^ strains, along with the isogenic WT (wild type) strain) or *ssb*B promoter (in all other mutant strains, along with the isogenic WT strain), as listed in Appendix A. Cultures were started by diluting the stock cultures 50- or 100-fold in C+Y medium adjusted to the desired pH and supplemented with luciferin in a 96-well white NBS micro plate (Corning) [17]. Relative luminescence units (RLU) and OD_492_ values were recorded every 5 to 10 min (depending on the experiment) during growth at 37 °C with no shaking in a Varioskan Flash luminometer (Thermo Fisher Scientific Waltham, MA, USA).

HPUra (6(*p*-Hydroxyphenylazo)-uracil), MMC (mitomycin-C) and IPTG were added to the growth medium at the final concentrations indicated in the figures. OD_492_ values and luminescence values reported as RLU/OD_492_ ratio were measured in real time over the growth of each strain under the different conditions described in the figures. Each figure reports a representative experiment or a mean (and standard deviation) of at least three independent experiments, which gave rise to similar results. 

For the plating tests, stock cultures were 10-fold serially diluted, spotted on CAT-agar [18] containing 2% horse blood, 500 units/mL of catalase and IPTG (as indicated), and incubated overnight at 37 °C. 

Genome modifications were performed by natural transformation with the use of chromosomal or PCR fragments as transforming DNA (tDNA), following the procedure described previously [19]. Briefly, freshly inoculated stock cells were treated at 37 °C for 10 min with synthetic Competence-Stimulating Peptide (CSP1; 100 ng mL^−1^) to induce competence for genetic transformation. Next, tDNA was added to the competent cells, which were further incubated for 20 min at 30 °C before plating on CAT-agar plates supplemented with 4% horse blood and, when required, with the appropriate concentration of the following antibiotics and after phenotypic expression for 120 min at 37 °C: kanamycin (Kn; 500 µg mL^−1^), spectinomycin (Spc; 200 µg mL^−1^), and chloramphenicol (Cm; 9 µg mL^−1^). Constructs obtained with PCR fragments as tDNA were checked by PCR and sequencing. 

### 2.2. Pneumococcal Strain Constructions

#### 2.2.1. *recA*, *recO*, *recR*, *recF*, and *rexB* Strains

The *recA^−^* R3309 strain was constructed by transformation of the R825 strain with genomic DNA of the R209 strain that contains the *recA*::*cm* allele. The *recO^−^* strain R2411 was constructed by transformation of the R895 strain with genomic DNA of the R2372 strain that contains the *recO*:: *spc^13C^* allele (Johnston et al. 2015). The *recR^−^* strain R2412 was constructed by transformation of the R895 strain with genomic DNA of the R2373 strain that contains the *recR*:: *kan^15C^* allele (Johnston et al. 2015). The *rexB^−^* strain R4036 was constructed by transformation of the R825 strain with genomic DNA of the DPH14 strain that contains the *rexB*:: *spc* allele [20].

#### 2.2.2. CEP-P*_lac_*::*yab*A Strain

The CEP-P*_lac_*::*yab*A strain (R3319) derives from the R895 and R3310 strains. The R3310 strain comprises the Chromosomal Expression Platform (CEP)-P*_lac_*, which is inserted at the *amiF* locus and is composed of a Kn-resistance gene, the *lacI* gene, the IPTG-inducible P*_lac_* promoter, and the *luc* coding sequence as previously described [21]. *The* CEP-P*lac*::*yab*A (R3319) strain was generated by transforming the R895 strain with a SOEing PCR product composed of the CEP-P*_lac_::yab*A sequence framed by the *amiF-lacI-P_lac_* and *kan-treP* sequences and obtained from the three PCR products: *amiF-lacI-P_lac_* (primers OVK134 and OVK169 using R3310 genomic DNA as a template), *yabA* (primers *OVK229* and OVK230 using R895 genomic DNA as a template), and *kan-treP* (primers OVK172 and OVK139 using R3310 genomic DNA as a template).

#### 2.2.3. CEP-P*_lac_*::*dna*A Strain

The CEP-P*_lac_*::*dna*A strain (R3318) was obtained by transformation of the R895 strain with plasmid pVK^*CEP-*P*lac*::*dna*A^, which includes the *lac*I*-*P*_lac_::dna*A*-kan* construct framed by *ami*F and *tre*P sequences constructed by the SOEing with the pUC19 plasmid backbone. This plasmid derives from the plasmid pVK^CEP-P*lac*::*I-sceI*^ constructed as follows: first, a CEP-P*_lac_*::I-*sce*I strain (R3312) was constructed by transformation of R825 with *a* SOEing-PCR fragment assembled from three PCR products corresponding to *amiF-lacI-P_lac_* (primers OVK134 and OVK169 and genomic DNA from R3310 as a template), I-sceI (primers OVK170 and OVK208 and plasmid pUC19-I-SceI as a template), and *kan-treP* (primers OVK172 and OVK139 using genomic DNA of R3310 as a template (the I-sceI coding sequence was synthesized with codons optimized for *S. pneumoniae* as defined by the OptimumGene™ algorithm and cloned into the pUC19 vector by Genscript USA to generate plasmid pUC19-I-*sce*I)*;* second, a PCR fragment corresponding to CEP-P*_lac_*::I-*sce*I (primers OVK301 and OVK304 using genomic DNA from R3313 strain) digested PstI/EcoRI was cloned into pUC19^(NdeI)^ (pUC19^(NdeI)^ is a pUC19 derivative mutated for its NdeI site); third, to generate the pVK^CEP-P*lac*::*dna*A^ plasmid, a PCR fragment encompassing *dna*A coding sequence, generated with OVK214 and OVK3 primers and using R895 as a template, was digested by NdeI and BamHI and substituted for the I-SceI coding sequence into pVK^CEP-P*lac*::I-*sce*I^ digested by NdeI and BamHI. 

##### *^e^*P*_lac_*::*dnaA*, ^e^P*_lac_*::*dnaE*, and ^e^P*_lac_*::*dnaC* Strains

To construct these strains, we first designed a construct allowing the targeted insertion of the P*_lac_* promoter at any endogenous locus of the pneumococcal chromosome and defined as ^e^P*_lac_* hereafter (with ^e^ indicating endogenous). This construct derived from the CEP-P*_lac_*::*luc* ectopic construct of the R3110 strain [21], with a Spc-resistance gene (*aad9*) substituting the Kn-resistance gene to give the R3311 strain. This substitution was achieved by transformation of the R3110 strain with a SOEing-PCR product between three PCR products corresponding to lacI (primers OVK136 and OVK213 using genomic DNA of R3310 as a template), *aad9* (primers OVK210 and OVK28 on plasmid pR412 [22], and *P_lac_::luc* (primers OVK214 and OVK215 using genomic DNA of R3310 as a template), as well as selecting for Spc^R^ Kn^S^ clones.

The ^e^P*_lac_*::*dna*A strain (R3317) was obtained by transformation of R895 with a SOEing PCR fragment generated from three overlapping PCR products made of the 5′-dnaA sequence (primers OVK196 and OVK218 using genomic DNA of R895 as a template), the *lacI* and P*_lac_* promoter sequences up to the ATG initiation codon (primers OVK136 and OVK169 using genomic DNA of R3311 as a template), and the *dnaA* coding sequence (primers OVK219 and OVK199 using genomic DNA of R895 as a template), as well as selection for Spc^R^ clones. Of note, the natural TTG start codon of *dnaA* has been maintained in this synthetic construct and is preceded by the ATG codon from the P*_lac_* promoter.

The *^e^*P*_lac_**::dnaC* (R3316) strain was obtained by transformation of R895 with a SOEing PCR product generated from three overlapping PCR products made of the 5′-*dnaC* sequence (primers OVK245 and OVK246 using genomic DNA of R895 as a template), the *lacI* and P*_lac_* promoter sequences up to the ATG initiation codon (primers OVK136 and OVK237 using genomic DNA of R3311 as a template), and the *dnaC* coding sequence (primers OVK247 and OVK248 using genomic DNA of R895 as a template). 

The ^e^P*_lac_**::dnaE* (R3315) strain was constructed by transformation of R895 with a SOEing-PCR product generated from three overlapping PCR products made of the 5′-*dnaE* sequence (primers OVK251 and OVK252 using genomic DNA of R895 as a template), the *lacI* and P*_lac_* promoter sequences up to the ATG initiation codon (primers OVK136 and OVK237 using genomic DNA of R3311 as a template), and the *dnaE* sequence (primers OVK256 and OVK254 using genomic DNA of R895 as a template). Of note, an extra sequence of seven nucleotides was found inserted between the Shine-Dalgarno and the ATG of the P_lac_ promoter in this synthetic ^e^P*_lac_**::dnaE* construct.

These Spc^R^ strains were selected on solid THY medium (Thodd Hewitt broth supplemented with 2% yeast extract (Difco)) supplemented by IPTG (10 μM). Next, clones were screened for their IPTG dependency for growth, and the genomic region encompassing the fused PCR fragments was fully sequenced. 

### 2.3. ori/ter Ratio Measurement

Genomic DNAs were prepared using a DNeasy DNA extraction kit (Qiagen). Quantitative realtime PCR was performed on a Realplex thermocycler device using Sybergreen dye (iQ SYBR Green Supermix, Biorad, Hercules, CA, USA) to amplify specific origin or terminus sequences. Oligonucleotides used for PCR amplification were chosen by using the ‘eprimer3′ program (http://bioweb.pasteur.fr (accessed on 27 Feburary 2013)). The amplified origin sequence corresponds to a 129-bp-long PCR product obtained by using primers OVK36 and OVK37 that target the 7594–7722 region of the *S. pneumoniae* chromosome (Appendix A). The terminus sequence is a 125-bp-long PCR product obtained by using primers OVK46 and OVK47 that amplify the 1,046,843–1,046,967 region of the *S. pneumoniae* chromosome (Appendix A). These two pairs of primers, defined as *ori* and *ter,* respectively, exhibited 98% amplification efficiency. The *ori*/*ter* ratio was measured at different time points of the cell growth as indicated in the text and/or in the legend of the figures. Each *ori*/*ter* measurement was performed using at least two independent cell cultures.

## 3. Results

### 3.1. RecA-Mediated Recombination Prevents Competence Development in S. pneumoniae

To compare competence development in wild-type (WT) and *rec*A mutant (*recA*^−^) strains, we monitored the expression of a competence-induced gene (*comC*) fused to the firefly luciferase gene, as reported previously ([17]; see Material and methods). In this assay, light emitted by the growing cells is recorded in real time as relative luminescence units (RLU) normalized to cell density expressed as OD_492_ (see Material and methods; Figure 1A). Under conditions appropriate for competence development, this gives rise to a characteristic, transient peak of competence gene expression during exponential growth. This expression, and competence development itself, declines progressively as the initial pH of the medium used is acidified (Appendix A; [23,24]). Conditions that stimulate or depress competence induction, such as certain mutations and drugs, can overcome the pH effect and thus cause the return or disappearance of the expression peak, changes termed hereafter competence-up (‘*cup*’) and competence-down (‘*cdo*’) phenotypes, respectively. We observed that a *rec*A- mutant exhibits a *cup* phenotype, as it develops spontaneous competence at a pH at which the WT strain does not (Figure 1B). This result implies that the RecA protein impedes pneumococcal competence development. 

As pneumococcal competence is known to be induced by various types of DNA damage [13,15], the *cup* phenotype of the *recA* strain could stem from types of damage that are no longer efficiently corrected by RecA recombination activity. RecA-mediated maintenance of genome integrity is known to proceed by distinct recombination pathways, triggered by distinct types of damage and involving specific sets of effectors [25,26,27]. Early effectors of these pathways promote the formation of the RecA presynaptic recombination filaments on ssDNA, upon which RecA then executes the DNA strand-exchange reactions that restore the genome. Two distinct recombination effectors are involved in genome maintenance in bacteria. One is the RecF–RecO–RecR triad that acts at DNA gaps [27]; the other is the RecBCD complex in *E. coli* [27], termed RexAB in *S. pneumoniae* [20], which acts at double-strand breaks (DSB). To evaluate the relative importance for competence of these two recombinational pathways in *S. pneumoniae*, we monitored competence induction in *recO*^−^, *rec*R^−^, and *rex*B^−^ mutants by recording the expression of a *luc* transcriptional fusion with the *ssb*B competence gene. Each mutant exhibited a *cup* phenotype (Figure 1B). These results show that impairment of recombinational pathways that repair the genome is a potential source of competence induction.

### 3.2. The Cup Phenotype of the recA^−^ Mutant Is Exacerbated by the DNA Damaging Agent Mytomycin-C in a Dose-Response Manner

To further verify that the *cup* phenotype of the *rec*A^−^ mutant results from a deficiency in repairing genome damage, we compared competence development of WT and *recA*^−^ strains in the presence of increasing amounts of MMC, a DNA damaging agent known to induce pneumococcal competence when added at concentrations impacting growth and cell viability [13]. Here, the pH of the medium was adjusted to prevent spontaneous competence development in *rec*A^−^ cells. Under these conditions, competence of the WT strain was readily induced by MMC at 30 ng·mL^−1^ and above (Figure 2). Competence was also induced by MMC in the *rec*A^−^ strain, but at much lower concentrations of the drug (Figure 2). Furthermore, the competence peak of the *rec*A^−^ strain occurred earlier during growth than for the WT strain at all MMC concentrations tested, concomitant with an increase in the induction level in proportion to MMC concentration. This result is likely to be a consequence of a gradual accumulation of unrepaired DNA lesions that foster competence development. In WT cells, most of these DNA lesions would be efficiently repaired by RecA recombination pathways, avoiding competence development until the extent of damage exceeds the capacity for efficient repair. Thus, these results support the notion that signaling leading to competence induction is a quantitative response, developing in proportion to the damage suffered by the DNA. 

### 3.3. Competence Induction by Replication Inhibitor HPUra Is Independent of Gene Dosage Changes

Several drugs that inhibit replication have been reported to induce competence in *Pneumococcus* [13,15]. Chromosome replication in *S. pneumoniae* is initiated bidirectionally from the origin, *ori*C, situated between the two closely linked competence control operons, *com*CDE and *com*AB, and terminates diametrically opposite in the *ter*C region (Appendix A). An analysis of transcription levels in cells treated with replication inhibitors suggested that the increased gene dosage of origin-proximal genes resulting from continued initiation on chromosomes with paused replication forks would raise the relative rate of *com*CDE gene expression enough to activate the ComABCDE positive feedback loop, leading to competence induction [15]. In this study, however, replication drugs were used at only one concentration, leaving open the possibility that other mechanisms might not have come to light. 

We explored competence development over a range of concentrations of the replication damaging agent, HPUra, a nucleotide derivative that specifically blocks PolC polymerase progression by competing with dGTP [28]. The addition of increasing amounts of HPUra to the WT strain gradually reduced the growth rate of WT cells (Figure 3A). We monitored competence development over this range of HPUra concentrations in a medium adjusted to a pH that does not allow spontaneous competence development without the drug. Figure 3A shows that *ssbB::luc* expression increased steadily with HPUra concentration up to 300 ng·mL^−1^, at which point the competence of the population was fully developed. We next determined the effect of HPUra concentration on relative *ori*C gene dosage, by qPCR measurement of the *ori*/*ter* ratio (see Appendix A and Section 2). Appendix A shows the time course of *ori*/*ter* response to addition of HPUra at 50 ng·mL^−1^ in comparison with the *ori*/ter ratio of cells grown without the drug. In the absence of HPUra, the *ori*/*ter* ratio remained constant at 1.6 (Appendix A), corresponding to the value obtained by sequencing of chromosomal DNA extracted from exponentially growing pneumococcal cells [29]. In the presence of HPUra, the *ori/ter* ratio gradually increased over time (Appendix A). Based on this calibration, we measured the *ori*/*ter* ratio over the whole range of HPUra concentration, in samples taken at the last time-point in the series (60 min). As shown in Figure 3B, the correlation between the *ori*/*ter* ratio and HPUra concentration broke down above 100 ng mL^−1^, whereas competence gene expression continued to rise. Indeed, at the highest HPUra concentration (1000 ng mL^−1^), the *ori/ter* ratio was nearly identical to that in the absence of drug (Figure 3B) despite inducing the highest level of competence (Figure 3A). 

In light of these results, we also measured the *ori/ter* ratio of WT cells grown in the presence of increasing amounts of MMC. In this case, the *ori/ter* ratio increased proportionally with the dose of MMC, with an inducible value of competence development above 40 ng·mL^−1^ (Appendix A). 

These results show that an increase in the *ori/ter* ratio resulting from replication impediments, even massive, is not necessarily correlated with induction of competence. This implies that increased *com* gene dosage at *ori*C is not a sufficient, or perhaps even necessary, signal for competence development. Rather, competence gene expression is readily induced at levels of HPUra that provoke replication fork arrest, as seen by the sharply decreased *ori/ter* ratio and of growth rate at the highest HPUra concentrations tested. These findings indicate that competence development is poised to respond to disruptions in DNA replication. We further explored this idea by examining the induction of competence in response to other essential replication effectors.

### 3.4. Modulation of Competence Development by Altering DnaA Concentration

DnaA is the widely conserved master regulator of chromosomal DNA replication initiation in bacteria. Its activity is regulated by several mechanisms, some species-specific, which combine to coordinate replication with the cell-cycle. One key parameter known to affect replication in *B. subtilis* is overexpression of *dnaA* from its native locus, which leads to DNA stresses and ends up in SOS induction [30]. 

To investigate whether artificial modulation of DnaA concentration in the cell could influence pneumococcal competence development, we replaced the promoter region of *dna*A with a synthetic IPTG-inducible P*lac* promoter (see Material and methods). Growth of the resulting ^e^P*lac*-*dna*A strain was IPTG-dependent, as shown by the spot test assay presented in Figure 4A. Cell growth was severely and progressively retarded at ≤2.5 μM IPTG, and scarcely any colony was detected with none. Above 5 μM IPTG, colonies were identical in shape and number to those of the WT, but at the highest IPTG concentration tested (100 μM), colonies appeared smaller, indicating that *dna*A overexpression is detrimental to the cell cycle. The measured *ori*/*ter* ratios were correlated with colony-forming ability: *ori*C was under-replicated below 2.5 µM IPTG, replicated at WT frequency at 5–10 µM, and over-replicated at 100 µM (Figure 4C). We then analyzed competence development in the ^e^P*lac*-*dna*A strain grown in liquid medium with this range of IPTG concentrations and at different pH values. Cells grown at the highest *dna*A expression level exhibited a *cup* phenotype in comparison with cells grown between 5 and 10 μM IPTG (Figure 4B). In contrast, when grown in competence-permissive medium to allow a more graded observation of competence response, the ^e^P*lac*-*dna*A strain developed competence later and less strongly at inducer concentrations below 2.5 µM than when induced at levels enabling normal growth (5–10 µM). We also noted that the X_A_ time taken to induce competence gradually decreased between 2.5 and 10 µM, while the *ori*/*ter* ratio was effectively constant (Figure 4B). These results point to a correlation between the *dna*A expression level and the tendency of cells to develop competence, independently of relative gene dosage. 

The *dna*A gene is the first gene of an operon whose second gene is *dna*N, which encodes the DNA polymerase processivity factor of the replisome [30]. Consequently, DnaN concentration will also be modulated by IPTG concentration in the ^e^P*lac*-*dnaA* strain, and this in turn could impact replication and the propensity of the cells to develop competence. Indeed, inhibition of DNA synthesis in *B. subtilis* by overproduction of DnaA was shown to result from DnaA-mediated repression of the *dna*A-*dna*N operon and the consequent depletion of the DnaN protein [30]. Accordingly, we performed the same experiment in *S. pneumoniae* to explore how over-expression of DnaA alone affects competence development. We inserted an IPTG-inducible P*lac*-*dna*A construct at an ectopic position on the chromosome (see Section 2). The resulting CEP-Pl*ac*-*dna*A strain was grown with 100 μM IPTG in a medium adjusted to a pH non-permissive for competence development without IPTG. As shown in Appendix A, DnaA over-expression readily induced competence and also hindered growth more severely than in the case of the ^e^P*lac*-*dna*A strain at the same IPTG concentration (see Figure 4B). Thus, increasing the concentration of DnaA itself induces competence development. 

We then investigated the *cdo* phenotype shown by the ^e^Plac-*dna*A strain upon growth without IPTG (Figure 4B). For this, we used a (*S. pneumoniae*) strain overproducing a close homologue of the *B. subtilis* YabA protein, which has been demonstrated to down-regulate replication initiation at *ori*C by interacting with DnaA [31,32,33]. Importantly, *B. subtilis* YabA has been shown not to interfere with transcriptional regulation by DnaA [34]. As shown in Appendix A, we found that IPTG-induced expression of pneumococcal YabA from an ectopic P*lac*-*yab*A construct led to a *cdo* phenotype. This result strongly supports the notion that a reduced rate of DnaA-mediated replication initiation at *ori*C depresses spontaneous competence development. This *cdo* phenotype could stem from diminished expression of the *com*CDE operon, resulting from reduction of the *ori*/*ter* ratio due to altered timing of replication from *ori*C. To explore this possibility, we determined whether ^e^P*lac*-*dna*A cells grown without IPTG could develop competence upon addition of synthetic CSP to the growth medium. As shown in Appendix A, CSP added at any time during growth led to immediate induction of competence in ^e^P*lac*-*dna*A cells cultivated without IPTG or with 5 μM IPTG (in a medium of pH not permitting spontaneous competence development in the latter case). This indicates that DnaA-depleted cells maintained ComD and ComE basal levels above the threshold needed for CSP sensing and competence induction.

Taken together, these results support the idea of a cause-and-effect relationship between DnaA concentration and spontaneous development of pneumococcal competence. They also indicate that the relationship is based on the rate of initiation from *ori*C rather than on the regulatory properties of DnaA.

### 3.5. Aborted Replicative DNA Synthesis Fosters Competence Development

The experiments until this point revealed that alterations in the active concentrations of PolC and DnaA influenced spontaneous competence development. Previous experiment with HPUra revealed that reducing the active concentration of PolC resulted in a *cup* phenotype, most probably as a consequence of replication fork arrest and/or collapse. To delve further into this correlation, we used the same approach to examine the consequences of varying availability of the replicative helicase DnaC [35] and the replisome polymerase DnaE [36,37], both being essential for cell viability. In *B. subtilis*, DnaC and DnaE intervene, in that order, prior to DnaN and PolC in the assembly of replication forks at *ori*C and in re-initiation at interrupted forks, respectively [38]. Thus, depletion of either enzyme is expected to severely impede replication, at the initiation and the elongation steps, in both cases leading to the accumulation of arrested forks. We constructed two strains to enable IPTG-dependent modulation of *dna*C and *dna*E expression at their native chromosomal loci. As expected, growth of these strains was IPTG-dependent (Figure 5A and Figure 6A). Notably, however, depletion of these proteins was not complete, as inferred from the increase in the cell density of cultures even without IPTG (Figure 5B and Figure 6B). Growth under these conditions below 1 μM IPTG was, however, not sufficient to generate visible colonies in solid medium (Figure 5A and Figure 6A). To measure spontaneous competence development in these strains over a range of IPTG concentrations, we used cells initially cultured at an IPTG concentration permitting WT growth rate and the *ori/ter* ratio and at a pH preventing induction of competence. Under these conditions, both the ^e^P*lac*-*dna*E and ^e^P*lac*-*dna*C strains exhibited a *cup* phenotype when grown below 1 μM IPTG (Figure 5B and Figure 6B). The *ori*/*ter* ratio increased slightly in the ^e^P*lac*-*dna*E strain under these conditions (Figure 5C). In stark contrast, the *ori*/*ter* ratio fell markedly in the ^e^P*lac*-*dna*C strain grown below 1 μM IPTG (Figure 6C), i.e., under the conditions leading to competence development. 

Taken together, these results reinforce the notion that replication fork arrest and collapse are major triggers of competence development in *S. pneumoniae*.

## 4. Discussion

In this study, we have investigated the causal relationship between damage to the integrity or replication of the genome and competence induction in *S. pneumoniae*. We observed that deficiencies in RecA-directed repair stimulated competence development and conclude that competence induction stems from the sensing of injuries that are continuously generated during growth but normally repaired. Interestingly, competence in *Helicobacter pylori*, recorded by transformation efficiency, was reported to be improved in DNA repair mutants [39]. Recombinational DNA repair pathways are known to buttress chromosomal DNA replication by promoting the rescue of damaged forks via mechanisms that differ according to the type of injury suffered [27,40]. Thus, the *cup* phenotype of the *rec*A, *rec*O, *rec*R, and *rex*A mutants would reflect their failure to efficiently repair damage at replication forks (Figure 1B). Supporting this proposal, a *cdo* phenotype appears upon inhibition of DnaA-mediated initiation (by overexpression of the YabA replication repressor; Appendix A): the lower initiation rate would reduce the frequency of forks and hence of fork failure and the signals triggering competence. Failure of repair was not the only source of damage signals: increasing damage frequency by addition of replication inhibitors, such as MMC and HPUra, also accelerated competence development. Common to all the disruptions of replication that lead to competence development are unrepaired replication forks.

### 4.1. How Does Replication Stress Induce Competence?

Previous studies that linked replication stress to pneumococcal competence induction used exogenous drugs to specifically alter either a replication protein, the dNTP pool, or DNA integrity [13,15]. Among these drugs, the most specific for the replication process is HPUra, which effectively reduces the concentration of active replisomal PolC DNA polymerase in a dose-response manner. Here, we constructed strains that express the DnaC helicase and the DnaE DNA polymerase of the replisome in proportion to the concentration of added IPTG. The diminution of the active pool of DnaC and of DnaE molecules led to the induction of competence, mirroring the effect of high concentrations of HPUra on WT cells (Figure 3, Figure 5 and Figure 6).

It was previously proposed that the replication defect caused by HPUra was converted into a competence-inducing signal as a consequence of the higher relative dosage of the *ori*-proximal *com*CDE operon resulting from deceleration of the replication forks. HPUra-treated cells were shown to express the *com*CDE operon at higher levels than untreated cells, and this was proposed to activate the ComABCDE positive feedback loop and turn on competence [15]. A central parameter of this mechanism is the *ori*/*ter* ratio, which should increase under replication stress and be correlated with competence induction. We found instances in which this correlation does not hold. An extreme case was that of the ^e^P*lac*-*dna*C strain, which exhibited a *cup* phenotype when grown with concentrations of IPTG at which the *ori*/*ter* ratio gradually declined (Figure 6). Indeed, the *ori*/*ter* ratio varied for each replication stress and was not correlated with the propensity of the cells to develop competence. Thus, a high *ori*/*ter* ratio, >3, was found with HPUra at 50 ng·mL^−1^, which, however, was not followed by competence induction (Figure 3). By contrast, almost no change in the *ori*/*ter* ratio was found for WT cells grown with 1000 ng·mL^−1^ HPUra (Figure 3) or for the ^e^P*lac*-*dna*E cells grown without IPTG (Figure 5), although competence was readily induced in both situations. Thus, rather than increased *com*CDE gene dosage and expression, the factor common to all situations of replication stress that foster competence induction was the arrest of replication forks. 

In several cases, the propensity of the cells to develop competence was found to be proportional to the intensity of the stress applied. The gradualness of the response was manifested by a change in timing of competence induction, i.e., the more stress, the shorter the period (X_A_) preceding the peak of expression of competence genes. This was observed in experiments with increasing concentrations of MMC applied to *recA*^−^ cells (Figure 2), and to ^e^P*lac*-*dna*E and ^e^P*lac*-*dna*C cells grown with 1 μM and 0 μM IPTG (Figure 5 and Figure 6). Importantly, this dose-response effect should be considered in the light of the two-phase development of competence in pneumococcal populations and the mechanism underlying its coordination between cells. We previously provided genetic evidence that competence of a pneumococcal population develops in a self-activating (autocrine) stage, followed by a wholesale propagation (paracrine) stage (Figure 1A; [9]). During the initial phase, there is a steady increase in the number of individual cells that undergo physiological stress of a kind that induces the ComABCDE feedback loop. When the fraction of such cells rises to a certain level, CSP-mediated induction of neighboring cells through cell-to-cell contact becomes frequent enough to thrust the population into the second phase where competence spreads throughout, as seen by a peak upon monitoring of competence gene expression. The proportional response of competence induction to replication stress reported here is readily explained in this scenario. Lowering IPTG concentrations in cultures of ^e^P*lac*-*dna*E and ^e^P*lac*-*dna*C strains would be expected to progressively reduce DnaE and DnaC levels and increase the frequency of fork stalling. The fraction of individual cells that switch to competence would thus rise faster, leading to the shorter X_A_ periods observed. Similarly, addition of increasing amounts of MMC to *recA*^−^ cells will raise the fraction of cells accumulating unrepaired damaged DNA, thus progressively reducing the period of competence development (Figure 2). Notably, this gradual response is not observed for WT cells treated with the same range of MMC concentrations (Figure 2). In this case, recombinational repair is fully active up to a limit of the MMC concentration above which competence is induced, reflecting a saturation of repair capacity and leading to a maximal X_A_ value. The same reasoning applies to ^e^P*lac*-*dna*A cells grown at low IPTG concentrations. A gradual reduction of the replication initiation rate at *ori*C resulted in an extended X_A_ period of spontaneous competence development (Figure 4). The same effect was obtained with WT cells by overexpression of the replication inhibitor YabA (Appendix A). In both cases, the gradual diminution of the fraction of cells undergoing active replication exerts a corresponding delay in competence development.

### 4.2. Comparison of Pneumococcal Competence with the SOS Response

Unlike many bacteria, *Pneumococcus* does not have an SOS system to respond to DNA damage. Competence, similar to the SOS response, involves the induction a large set of genes that confer altered properties on the cell, but except for a small subset of proteins for DNA repair and recombination, there is little similarity in the competence and SOS regulons. Indeed, *rec*A is the only gene induced in both systems. Nevertheless, our study of competence development in response to DNA damage reinforces the notion that competence is the functional equivalent of SOS. Both systems induce a cell division inhibitor that provides a delay to enable cells to repair DNA breaks before division. In the case of the *Pneumococcus*, the ComM division inhibitor induced during competence was shown to be crucial for maintaining genome integrity during natural transformation [41]. Competence for transformation itself has been reported to improve survival to exposure to MMC [42], underlining the functional parallel of cell rescue between competence and SOS.

### 4.3. Concluding Remarks

This study focuses on the relationship between the induction of competence in *Pneumococcus* by replication stress. The underlying signaling mechanism appears not to depend on an increase in the copy number of the *ori*C-proximal *com*CDE operon following replication slowdown, as proposed previously [15]. This leaves open the question of the molecular mechanism that links genome damage to the induction of pneumococcal competence. Aborted replication forks appear to be essential and presumably trigger the signaling pathway that ultimately induces the ComABCDE positive feedback loop. Competence was shown also to be induced in response to genome damage in two other naturally transformable species, *Legionella pneumophila* and *Helicobacter pylori*, neither of them encoding a LexA homologue or possessing an SOS system [14]. The regulatory circuit controlling competence in these species is poorly defined, but distinct from the pneumococcal one. Thus, it will be interesting to establish whether a common mechanism leads to competence development in response to genome stress in these species regardless of the particulars of the pathway controlling competence gene expression. 

## Figures and Tables

**Figure 1 cells-10-01938-f001:**
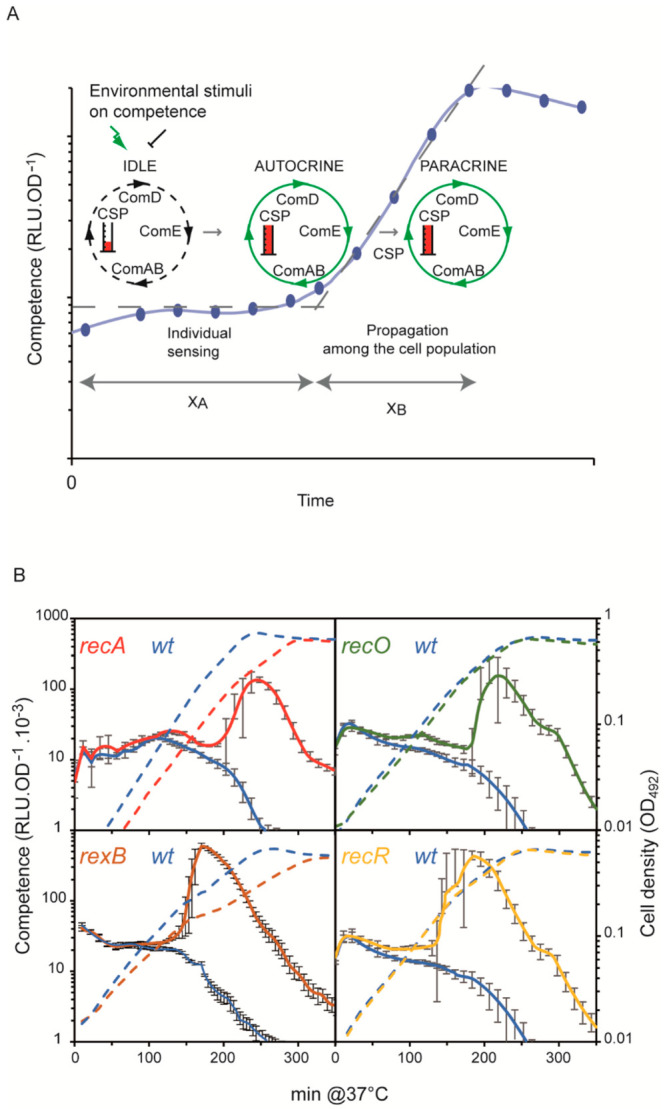
Pneumococcal competence development cycle: effect of repair–deficiency mutations. (**A**): Model of competence development in *S. pneumoniae*. The blue dots/blue line shows the development of competence in an exponentially growing culture measured by a transcriptional fusion of the luciferase gene in specific activity (Relative luminescence units (RLU) divided by the cell density (OD)); the growth curve is omitted for clarity. The core sensing and regulatory machinery of competence are composed of the *comA*, *comB, comC* (CSP), *comD,* and *comE* gene products, which define a positive feedback loop referred to as ComABCDE. The green arrow and the black T-bar represent external positive and negative inputs, which balance the idling of the core sensing machinery. At some point, growth conditions will provide enough positive input to activate the ComABCDE positive feedback loop in an autocrine mode in some of the cells. Next, this fraction of cells propagates competence within the population in a paracrine mode by transmitting the CSP via cell-to-cell contact. In this model, the autocrine and paracrine modes of competence development proceed in two distinct periods defined as X_A_ and X_B_, respectively [9]. X_B_ is followed by shut-off of competence gene expression. The dashed grey lines represent competence gene expression during the X_A_ and X_B_ periods. (**B**): RecA recombination repair pathways prevent competence development. Cells were grown in C+Y medium with initial pH adjusted with HCl to inhibit competence development of the reference strains R825 or R895 (blue curves); see also Appendix A. Competence of the population is shown as solid lines (RLU.OD_492_^−1^ × 10^3^) and cell density as dashed lines (OD_492_). Each panel shows a comparison of competence development of the WT strain with that of the mutant strains *rec*A^−^ (R3309; red), *rex*B^−^ (R4036; light brown), *rec*O^−^ (R2411; dark green), and *rec*R^−^ (R2412; orange) with that of their respective isogenic WT (wild type) strain (R825 for *rec*A^−^ and *rex*B^−^ strains, and R895 for the others). Data are the means of at least five independent experiments, with standard deviations shown for the RLU.OD_492_^−1^ measurements.

**Figure 2 cells-10-01938-f002:**
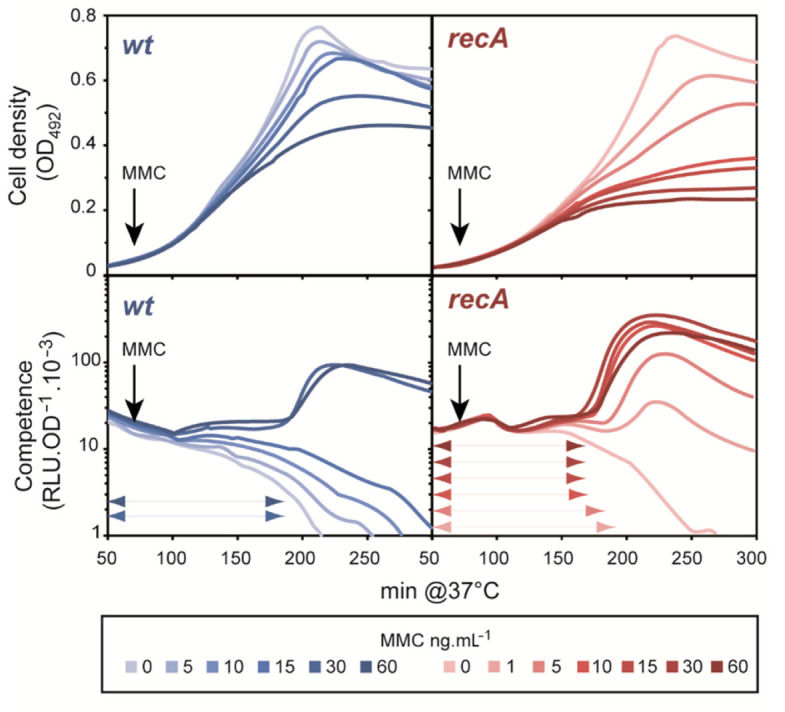
The lack of RecA exacerbates MMC-mediated competence induction. WT and *rec*A*^−^* cells were cultivated in C+Y medium adjusted to a pH non-permissive for spontaneous competence development. Cell density (OD_492_) is shown in the upper panels and competence gene expression (RLU.OD_492_^−1^ × 10^3^) in the lower panels recorded every 10 min. After 70 min of growth, MMC was added at the indicated concentrations, shown as light grey-blue to dark grey-blue for WT cells and light brown to dark brown for *recA^−^* cells. The curves represent the mean of two replicates. Standard deviations are omitted for clarity. Double arrowheads depict the X_A_ period.

**Figure 3 cells-10-01938-f003:**
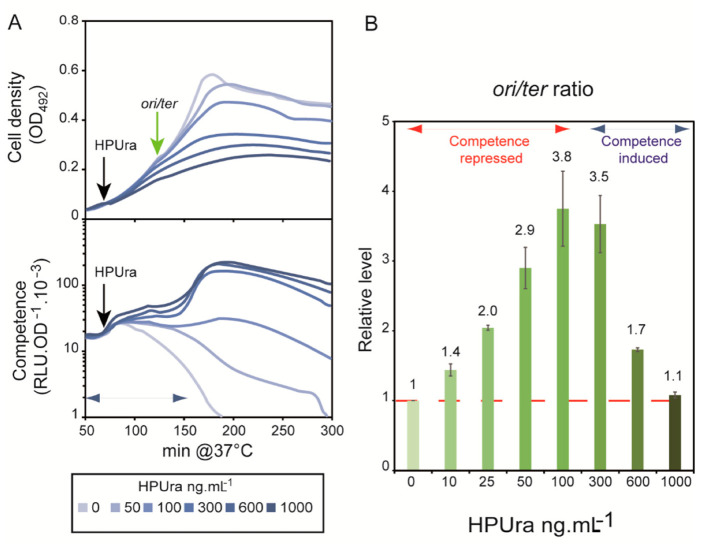
HPUra induces competence independently of its effect on the *ori*/*ter* ratio. (**A**): R895 cells were grown at a pH non-permissive for spontaneous development of competence. After 70 min of growth, HPUra was added at 0, 50, 100, 300, 600, and 1000 ng mL^−1^, light grey-blue to dark grey-blue, respectively. Cell density and competence plots are as in Figure 2. Curves represent the mean of three replicates for each condition. Standard deviations are omitted for clarity. (**B**): *ori*/*ter* ratios measured by qPCR on total DNA extracted from the WT R825 strain 1 h after HPUra addition at 0, 10, 25, 50, 100, 300, 600, and 1000 ng mL^−1^ (shown as light green to dark green respectively; see also Appendix A). The *ori*/*ter* ratio obtained for cells grown without HPUra was set to 1 and used as the reference for *ori*/*ter* ratios of cells at each HPUra concentration. The competence development status of each assay is indicated by the blue (induced) and red (basal level) double arrowheads. The means and standard deviations were derived from two replicates.

**Figure 4 cells-10-01938-f004:**
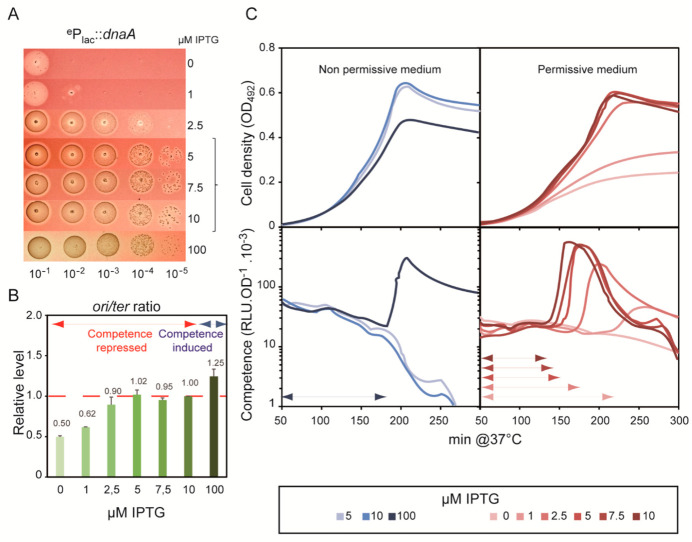
Differential control of competence development by modulated *dna**A* expression. In strain R3317, the endogenous *dnaA* promoter has been replaced by the inducible P*_lac_* promoter (^e^P*_lac_*::*dnaA*, see Section 2). (**A**): Growth assay. R3317 cells were grown to OD550 = 0.3 and were serially diluted and spotted on plates containing the indicated IPTG concentrations. The bracket covers IPTG concentrations enabling the mutant to mimic WT growth. (**B**): *ori*/*ter* ratios measured by qPCR on total DNA extracts from the R3317 strain (^e^P*_lac_*::*dnaA*) cultivated at the indicated IPTG concentrations and normalized to the *ori*/*ter* ratio of cells grown with 10 µM IPTG. The latter ratio also corresponds to the *ori*/*ter* ratio of the isogenic WT strain (red horizontal dotted line). The competence development status of each assay is indicated by the blue (induced) and red (repressed) double arrowheads (see Material and methods). The means and standard deviations were based on two replicates. (**C**): Growth (upper graphs) and competence gene expression (lower graphs) of the ^e^P*_lac_*::*dnaA* strain were monitored in media non-permissive (left panels) or permissive (right panels) for competence development of this strain grown at 10 mM IPTG. Double arrowheads specify the X_A_ period. Curves represent the mean of two replicates. Standard deviations are omitted for clarity.

**Figure 5 cells-10-01938-f005:**
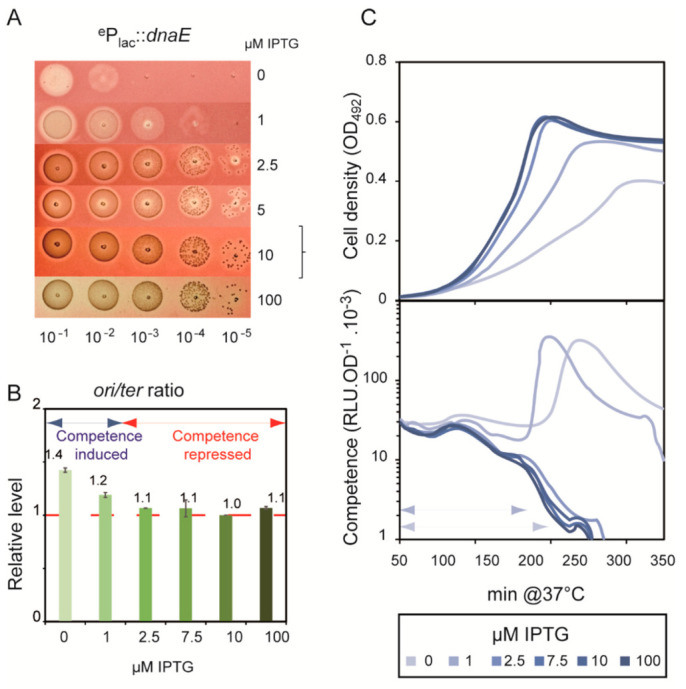
Reduction of *dnaE* expression induces competence. In strain R3315, the endogenous *dnaE* promoter has been replaced by the inducible P*_lac_* promoter (^e^P*_lac_*::*dnaE*, see Section 2). (**A**): Growth assay. R3315 cells were grown to OD_550_ = 0.3 and then diluted and spotted on medium containing the indicated IPTG concentrations. The bracket covers IPTG concentrations enabling the mutant to mimic *wt* growth. (**B**): *ori*/*ter* ratios measured by qPCR on total DNA extracted from the R3315 strain (^e^P*_lac_*::*dnaE*) cultivated at the indicated IPTG concentrations. The *ori*/*ter* ratio obtained for cells grown with 10 µM IPTG was used as the reference ratio to calculate the relative level (y-axis); it corresponds to the *ori*/*ter* ratio of the isogenic WT strain (red horizontal dotted line). The relative *ori*/*ter* ratio at each IPTG concentration is indicated, with standard deviations: values are the means of two replicates. The competence development status of each assay is indicated by the blue (induced) and red (repressed) double arrows. (**C**): Competence expression of the *ssbB::luc* transcriptional fusion in R3315 was monitored in C+Y medium adjusted at an initial pH non permissive for competence development when cells are cultivated with 10 µM IPTG, which fully complemented growth as WT cells. Curves represent the mean of four replicates for each growth condition performed at the indicated IPTG values. Standard deviations are omitted for clarity. Double arrowheads specify the X_A_ period.

**Figure 6 cells-10-01938-f006:**
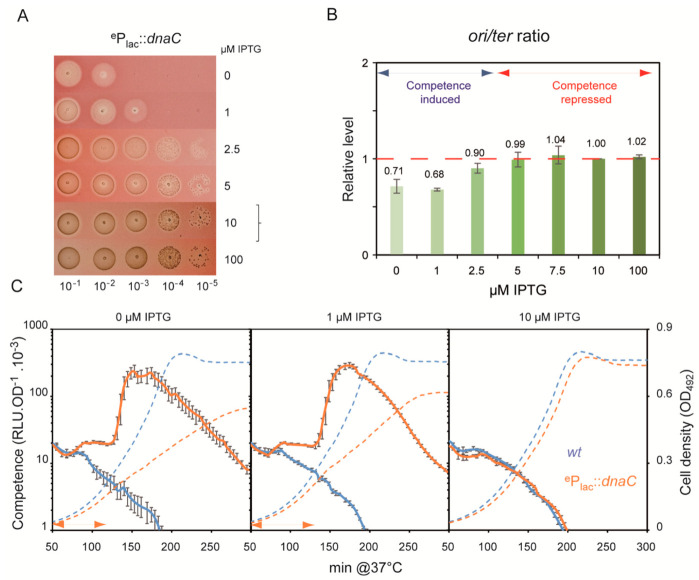
Reduction of *dnaC* expression induces competence and decreases the *ori*/*ter* ratio. In strain R3316, the endogenous *dnaC* promoter has been replaced by the inducible P*_lac_* promoter (^e^P*_lac_*::*dnaC*, see Section 2). (**A**): Growth assay. R3316 cells were serially diluted and spotted on medium containing the indicated IPTG concentrations. The bracket covers IPTG concentrations enabling the mutant to mimic WT growth. (**B**): *ori*/*ter* ratios measured by qPCR on total DNA extracted from the R3316 strain (^e^P*_lac_*::*dnaC*) cultivated at the indicated IPTG concentrations. The *ori*/*ter* ratio obtained for cells grown with 10 µM IPTG was arbitrarily set to 1 and used as a reference ratio (relative level), which corresponds to the *ori*/*ter* ratio of the isogenic WT strain (red horizontal dotted line). The relative *ori*/*ter* ratio measured in each condition is indicated. The competence development status of each assay is indicated by the blue (induced) and red (repressed) double arrows lines. The mean and standard deviation were generated from two replicates. (**C**): Expression of the *ssbB*::*luc* transcriptional fusion in the WT strain R895 (blue) and its isogenic ^e^P*_lac_*::*dnaC* strain R3316 (orange). Cell density (OD_492_) is presented as dotted lines and competence (RLU.OD_492_^−1^ × 10^3^) as plain lines recorded every 5 min. Cells were cultivated at 37 °C in C+Y medium with 0, 5 or 10 μM IPTG at a pH non-permissive for spontaneous development of competence for R3316 grown with 10 μM IPTG (as well as for the isogenic WT strain). Double arrowheads specify the X_A_ period.

## Data Availability

All data supporting the findings of this study are available within the article or from the corresponding author upon reasonable request.

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
