# Peer review of "Tight Interplay between Replication Stress and Competence Induction in Streptococcus pneumoniae"

_cells, 2021, doi:10.3390/cells10081938_

Round 1

Reviewer 1 Report

General comments

Khemicci et al  have investigated the causal relationship between a growth arrest (by DNA damage, its deficiency in repair or impairment of DNA replication) and competence induction in S. pneumoniae. However, competence-induced transient growth arrest has been observed only for S. pneumoniae and B. subtilis, as earlier documented by Oggiani´s (2004) and Dubnau´s (2001) groups.

The hypothesis that recombinational repair removes the competence induction signals can be misinterpreted, as supportive of “competence as a source of food” hypothesis. I consider that the authors clearly show that a replicative stress stimulates competence development and growth arrest rather than a SOS-like response. I consider that the work requires some fine tuning to clarify the presentation and avoid misinterpretations before acceptance for publication (see specific comments).

Specific comments

  1. L 17-19. RecA requires mediators to be loaded on the ssDNA, and in the absence of mediators the SOS induction is reduced and delayed. I guess that RexAB is not a mediator. Please rephrase.

  1. L 19. It is proposed that the absence of a recombination function, which is essential for homology search and pairing of the incoming ssDNA with its homologous complementary, stimulates the induction of a very expensive cellular stage (expression of >100 genes is altered), but the integrated ssDNA cannot be processed in the recA context, and the ssDNA is degraded. Are the authors providing a support for “competence as a source of food hypothesis”?

  1. L51-53. The authors should state the gene that codes for the peptide pheromone.

  1. L 253-256. I understand that absence of the recombinase (recA), RecA mediators (recO or recR) or end resection (rexB) function stimulates spontaneous competence induction in the absence of any pH adjustment. The WT control, in the absence of pH adjustment, is missing.

  1. L 261-262. Are the authors providing a support for “competence as a template for DNA repair hypothesis”? The authors propose that “stimulation of competence by recA strain could stem from types of damage that are no longer efficiently corrected by RecA recombination activity”. The acquisition of DNA template is minimal during transformation, <5% of total chromosomal.

  1. L 268-269. Is RexAB a RecA mediator? At least B. subtilis AddAB (counterpart of RexAB and RecBCD) does not work as a mediator (see doi: 10.1093/nar/gkv545). Furthermore, rexB cells are moderately sensitive to DNA damage, whereas recA extremely sensitive, suggesting that the division of labour observed in E. coli does not apply to bacteria of the Firmicutes Phylum (Happern et al 2004; doi: 10.1128/JB.188.2.353-360.2006).

  1. L 283-286. In Fig. 1 is shown that in the absence of recombinational repair, the unrepaired endogenous DNA damage accumulated triggers spontaneous competence induction. In Fig. 2 is shown that a lower dose of exogenous DNA damage is required to induce competence development upon pH adjustment in recA cells when compared to the wt control. These experiments are not supportive of the notion that competence is the functional equivalent of SOS response.

  1. L 355-356. In lane 285 is stated that “competence of the WT strain was readily induced by MMC at 30 ng.ml-1” and by ori/Ter >2, and in Fig S2 is shown that “competence of the WT strain was readily induced by ori/Ter at 1.9 and above”. I am confused by the statement that “an increase in the ori/ter ratio resulting from replication impediments . . . is not necessarily correlated with induction of competence”. I have missed the source of such conclusion. Please rephrase.

  1. L 384. what is the non-permissive medium? Is the C+Y medium adjusted to a low pH to avoid spontaneous competence induction?

  1. L 449-450. The available information as DnaE been an elongating polymerase is not correct. It was previously shown that PolC cannot use an RNA primer, but PolC can initiate replication and can elongate a hybrid RNA-DNA primer synthesized by DnaG and DnaE in concert (Sanders et al 2010; doi: 10.1093/nar/gkx493). Perhaps, DnaE should be considered as part of the primosome, together with DnaC and DnaG.

  1. L580-582. I am confused, induction of the SOS response requires RecA nucleated on ssDNA. Here, it is shown that the unrepaired DNA lesions accumulated in the recA context spontaneously induce competence development. I consider that competence development as a functional equivalent of SOS response can be ruled out from the data presented in Figures 1 and 2. I am not familiar with the cell division inhibitor induced upon competence, but I am confident that is clearly different from the SOS response cell division inhibitor. Finally, competence-induced transient growth arrest render cells refractory to antibiotic- and MMC-mediated killing and it has been postulated that this is because cells are not growing (see doi: 10.1534/genetics.108.099523).

Miscellaneous

  1. The name of the first author has a font size problem. In the Citation section the name of the middle author is missing.
  2. L 41. There is a fond size problem with the word “competence”
  3. L 53. TCS is described once, and its abbreviation is not described.
  4. L 60. What is the difference between Xa period and the former eclipse time?
  5. L 134. Please describe the abbreviation HPUra (6(p-Hydroxyphenylazo)-uracil). In the abstract is written HPUra, but in other part of the text mistyped as HpUra. The abbreviation of MMC is defined later (in L 282), but should be defined the first time that is used.
  6. L 193. Please remove the underline and place it in italic.
  7. L 683-688. The same reference has two different entries (2014a & 2014b)

Author Response

Please see the attachment. Our point by point responses are written in green.

Reviewer 2 Report

Khemicci et al focus on the relationship between progression of competence and replication stress in Pneumococcus. The underlying signaling mechanism appears not to depend on an increase in the copy number of the oriC-proximal comCDE operon following replication slowdown, as proposed previously. Transforming DNA requires protection from endogenous DNase(s) immediately after uptake by S. pneumoniae competent cells. Whereas in wild-type cells about 25% of incoming ssDNA is processed into recombinants, absence of RecA resulted in incoming ssDNA being very rapidly degraded. The authors provide an explanation for this and why they think that competence increases. Experiments are presented in a logical manner, and are mostly well controlled. While the work does not start with a clear working hypothesis, this develops within the results section, and can be easily followed.

Major points

  1. A) I have a conceptual problem with the term “competence induction”, or “competence development”. As I understand, competence peaks during mid exponential growth, and then declines. All effects observed in mutant cells or after addition of HPura do not change the level of competence induction seen by the Luciferase assays, but cause a derepression during transition into stationary phase. Maybe I am misinterpreting the results, but it looks to me as if e.g. lack of RecA leads to an extension of the window of competence, and reaching of higher levels during stationary growth. Thus, I can not follow the argument that RecA has an effect on the regulation of competence (during its normal timing), rather than changing the progression of competence program.

For example, line 258: “This result implies that RecA protein impedes pneumococcal competence development”. I can see normal competence levels during exponential growth, if lack of RecA had a direct effect on competence development, this should be seen or analysed during peak expression of regular competence. My hypothesis is that lack of RecA triggers an additional response, which in turn boosts competence. I would like to point out that this is quite a distinct interpretation of the data, so if the authors want to keep their claims on direct effects of competence INDUCTION, they need to perform different experiments, showing that higher expression levels of competence proteins are reached during the normal window of opportunity.

Along the same lines: line 276, 499

  1. B) legend to Fig. 2: “RecA suppression of competence induction by Mitomycin C.“ I think this is really a wrong use of the term “suppression”. What the lack of RecA does is it increases the effect caused by addition of MMC, but it does not reverse a negative effect. I would strongly suggest to keep the term “suppression” in the context of gene deletions affecting phenotypes of other gene deletions. In any event, I think that according to what is said above, the experiments shown do not show an effect on competence induction, but suggest that a lack of HR exacerbates effects on late competence caused by induction of DNA damage.

  1. C) line 442: „ In both cases, a cup 442 phenotype was observed under conditions leading to replication fork arrest and/or collapse.” This is only true for PolC inhibition, not for DnaA depletion, which affects initiation but not progression of replication. Please adjust your conclusions.

Minor comments.

Line 286: “by MMC at 40 ng.ml-1 and above“ there is no 40 ng data point in the figure.

Line 367: “One key parameter is 367 the concentration of DnaA molecules, alteration of which leads to replication stress 368 through over- or under-initiation at oriC“. It has been shown by< several group that altering DnaA levels two fold up or down does not affect timing, so the statement is too general.

Line 134, 318, 320, 331, 334, 336, 340, 347, 359, 360, 511, 524, 527, 538 should be “HPUra”

Line 119 Please reference what is mean with “C+Y medium”

Figure 3 change in axes and legends, should be “HPUra”

Line 18: „in contrast to SOS“; do you mean in contrast to SOS response?

Line 19: ‘do also’, grammatically incorrect, “do as well” would be ok, but “confer a similar effect” or something along those line better English

Line 41: formatting

Line 51: After two-component system should be (TCS) because abbreviation TCS was never introduced before

Line 58: What induces competence in Streptococcus? A more detailed description would be interesting. It is described later on that differences in pH will lead to competence induction. This should be explained in the introduction to not repeat it every time.

Later on you use comC as a reporter for competence. What is comC doing?

Line 60: bracket missing

Line 78: ‘and’ not italic

Line 81: loopin = loop in

Figure 1A: Relative luminescence units (RLU); please state in the description that the blue dots/blue line is the competence measured as relative luminescence units

Figure 1B: In different panels it seems like a different amount of data points were used. Why? Sometimes the standard deviations were high. Is it still comparable? Moreover, the grey lines of the standard deviations are not easy to relate to the respective curve in areas or curve overlapping. Why different reference strains were used? Did you also use a positive control for the experiment, e.g., a strain which is getting competent? Decimal separator for values on the right y-axis should be a point instead of a comma. Highlight in each panel that you compare the wildtype strain with an isogenic mutant. This is not clear from the labelling in the panels.

Line 111-241: Sometimes the font was cursive. Uniformity with formations of, e.g., headlines.

Line 119: What is C+Y media? What are the ingredients?

Line 120ff: why are you using different wavelength for measuring optical density of cell cultures? 550 vs 492 nm

Line 123: It was described that 10 µM IPTG was used, which is the concentration that results in a WT growth rate and ori/ter ratio value. As an evidence it would be useful to have the data of the wild type in the supplements or a reference.

Line 129: folds = fold

Line 131: how often were OD and RLU measured? Have they been incubated with constant shaking?

Line 134: usually referred to as ‘HPUra’

Line 140: is CAT agar a common thing? If not: what is it made of?

Line 191: Check font of restriction enzymes.

Line 225: What are the ingredients of THY medium?

Line 246: bracket missing

Line 247: referring to the methods section with the abbreviation ‘M&M’ seems colloquial

Line 251: why does the decline of pH lead to a decline of competence on a genetic/molecular level?

Line 283: Was MMC sub-inhibitory for the growth in your experiment?

Figure 2: Decimal separator for values on the y-axis should be a point instead of a comma. Box says ‘MC’ instead of ‘MMC’. Why don’t you see competence induction for the recA strain without MMC? Isn’t that the same condition like shown in Figure 1B where you observed competence signal?

Line 318: ‘whose’? HPUra is the derivative itself

Line 323: did it really increase steadily? There might be an effect for 100 ng/ml but is it really competence in the case of 50? How do you decide if they are really competent or the gene expression is only slightly increased? Is there a threshold? Or do you assume competence as soon as the curve differs from the one without HPUra addition?

Figure 3: Decimal separator for values on the y-axis should be a point instead of a comma. Box says ‘HpUra’ instead of ‘HPUra’.

Line 342: µM or ng/ml? Even if it was the same, make it uniform.

Line 345: why are you using two different strains for the growth/competence analyses and the ori/termeasurements?

Figure 4/5/6A: 0 and 1 µM spots barely visible

In general, it would be good if authors kept to shorter and precise sentences rather than to very long and complicated ones.

Author Response

(The authors gave the same response as above.)

Reviewer 3 Report

Khemicci et al. studied the mechanisms underlying the induction of competence in S. pneumoniae following a genomic stress. To do so they analysed the expression of the competence -induced gene comC in strains in which different DNA repair or replication genes were inactivated or in strains exposed to a DNA damaging agent. They observe that in the absence of DNA strand break repair (recA, recO, recR, rexB) competence is induced (cup phenotype) as it is in response to mitomycin C. The use of a replication blocking nucleotide analogue at variable concentrations showed that induction of competence does not correlate with the comCDE gene dosage increase which has previously been proposed to be the cause of competence induction following the perturbation of the genome replication. By modulating the levels of the replication proteins DnaA, DnaC or DnaE they show that the rate of replication initiation determines the induction of competence, independently of the gene dosage in the ori region. In conclusion, this manuscript presents a set of elegant and convincing experiments that challenges the previous hypothesis on the mechanisms of induction of competence in the pneumococcus by DNA replication perturbations.  

Minor points:

  • The sentence on lines 272-273 should be: “To evaluate the relative importance for competence of these two…”
  • References 34 and 35 are the same
  • Sentence on lines 597 to 600: A reference should be added for the pylori induction of competence in DNA repair mutants (i.e. Marsin et al., 2010, FEMS Microbiol Lett.)
  • The text requires minor editing. Some examples:

Line 81: loop in

Line 114: which derives

Line 375: identical in shape

Line 390: independently

Author Response

(The authors gave the same response as above.)

Round 2

Reviewer 2 Report

While the authors have answered my question about "competence induction", this may not be clear to a reader with the same question. I would encourage the authors - maybe at the proof stage - to incorporate a sentence to the effect " ... the term "competence induction" is simply to refer to the appearance of the peak of luciferase expression within the cell culture, always in comparison with a WT or unstressed situation" such that this point is defined once.